# Cell-Free Mitochondrial DNA: An Upcoming Non-Invasive Tool for Diagnosis of BK Polyomavirus-Associated Nephropathy

**DOI:** 10.3390/biom14030348

**Published:** 2024-03-14

**Authors:** Luying Guo, Sulin Luo, Xingxia Wang, Nengbo Zhang, Yamei Cheng, Jia Shen, Jianghua Chen, Rending Wang

**Affiliations:** 1Kidney Disease Center, The First Affiliated Hospital, School of Medicine, Zhejiang University, Hangzhou 310003, China; guoluying@zju.edu.cn (L.G.); 22318405@zju.edu.cn (S.L.); wangxingxia@zju.edu.cn (X.W.); 18814894538@163.com (N.Z.); 22018363@zju.edu.cn (Y.C.); jiashen@zju.edu.cn (J.S.); chenjianghua@zju.edu.cn (J.C.); 2Key Laboratory of Kidney Disease Prevention and Control Technology, Hangzhou 310003, China; 3National Key Clinical Department of Kidney Diseases, Hangzhou 310003, China; 4Institute of Nephrology, Zhejiang University, Hangzhou 310003, China; 5Zhejiang Clinical Research Center of Kidney and Urinary System Disease, Hangzhou 310003, China; 6Department of Nephrology, 903rd Hospital of PLA, Hangzhou 310013, China

**Keywords:** BK polyomavirus-associated nephropathy, mitochondrial damage, cell-free mitochondria DNA

## Abstract

Mitochondria are essential organelles that possess their own DNA. Mitochondrial dysfunction has been revealed in many kidney diseases, including BK polyomavirus-associated nephropathy (BKPyVAN). In this study, we introduce an innovative approach for non-invasive monitoring of mitochondrial impairment through urinary donor-derived cell-free mitochondrial DNA (ddcfmtDNA), addressing the crucial challenge of BKPyVAN diagnosis. Urinary samples were collected at the time of biopsy from a total of 60 kidney transplant recipients, comprising 12 with stable function, 22 with T cell-mediated rejection, and 21 with biopsy-proven BKPyVAN. Our findings reveal that the ddcfmtDNA-to-ddcfDNA ratio exhibits superior capability in distinguishing BKPyVAN from other conditions, with a cutoff value of 4.96% (area under curve = 0.933; sensitivity: 71.4%; and specificity: 97.1%). Notably, an elevation of ddcfmtDNA levels is associated with mitochondrial damage, as visualized through electron microscopy. These results underscore the promise of non-invasive monitoring for detecting subtle mitochondrial damage and its potential utility in BKPyVAN diagnosis. Further investigations are required to advance this field of research.

## 1. Introduction

Mitochondria, often described as the powerhouses of cellular respiration, are unique organelles widely believed to be descendants of an ancient Gram-negative bacterium that underwent an endosymbiotic process, giving rise to the progenitors of eukaryotic cells [1]. Mitochondria have their own mitochondrial (mt) DNA, consisting of a single double-stranded circular loop. The mtDNA contains 37 vital genes encoding proteins involved in processes such as oxidative phosphorylation, in addition to encoding transfer RNA and ribosomal RNA. Notably, mitochondrial damage has been implicated in various types of diseases. As a result, mtDNA can be released into circulation, potentially triggering a proinflammatory response [2,3] or serving as a biomarker of mitochondrial damage. Several investigations have reported that viruses employ their own encoded proteins to anchor mitochondria, aiming to exploit their host environment to its fullest during viral infection and replication.

BK polyomavirus, known to persist latently in the urothelial cells of immunocompetent individuals, is responsible for prevalent infections related to kidney transplantation [4]. A recent study utilizing single-cell RNA sequencing revealed that pathways resulting in mitochondrial dysfunction are upregulated in polyomavirus infection [5]. Specifically, the alterations in mitochondrial functions were associated with the expression of agnoprotein. Manzetti et al. have also shown that BK polyomavirus utilizes agnoprotein for mitochondrial targeting and network disruption to facilitate its replication, both in terms of rate and magnitude [6]. These compelling pieces of evidence suggest a new avenue for monitoring mitochondrial damage in the detection of BK polyomavirus-associated nephropathy (BKPyVAN) in kidney transplantations.

The advent of DNA sequencing technology has revolutionized the field of biomedical research, enabling the detection of cell-free DNA (cfDNA) in both plasma and urine. This breakthrough has significantly enhanced the convenience of monitoring allograft health status. In our recent report, we highlighted the role of urinary donor-derived cfDNA (ddcfDNA) in the non-invasive diagnosis of BKPyVAN in kidney transplant recipients [7]. ddcfDNA can be categorized into nuclear and mitochondrial DNA based on its origin. In this study, we introduce a novel method for measuring urinary cell-free mitochondrial DNA (cfmtDNA) in kidney transplant recipients (Figure 1) and address important questions related to the use of urinary cfmtDNA in the diagnosis of BKPyVAN.

## 2. Materials and Methods

### 2.1. Patients

This study was approved by the Research Ethics Committee of the First Affiliated Hospital of School of Medicine, Zhejiang University (Hangzhou, China; Approval Number 2017-642-1), and adheres to the principles of the Helsinki Declaration (2000). Between June 2017 and September 2019, a total of 55 kidney transplant recipients with allograft biopsy were enrolled in the study, consisting of 12 patients with stable graft function (STA), 22 patients with biopsy-proven T cell-mediated rejection (TCMR), and 21 patients with biopsy-proven BKPyVAN based on histopathology. The inclusion and exclusion criteria are detailed in our prior investigation [7]. Written informed consent was obtained from all patients before the commencement of the study.

### 2.2. Single Nucleotide Polymorphism (SNP) Site Selection and Probe Synthesis

SNP sites were selected from dbSNP (https://www.ncbi.nlm.nih.gov/snp/ (accessed on 26 May 2020)) and the Chinese population database (maintained in AlloDx Biotech, Co., Ltd., Shanghai, China), which is a catalog of genetic polymorphism sites constructed based on the whole-genome sample information of 5200 Chinese individuals. In total, 454 target SNP loci were obtained by screening for sites with minor allele frequencies (MAFs) of close to 0.45–0.55. Based on the genomic coordinates of these 454 target SNP sites, we designed upstream and downstream probes by extending 80bp nucleotide sequences from the SNP site coordinates for both the positive and negative strands, excluding the SNP site sequences. Bidirectional coverage detection of each SNP site was achieved by using both upstream and downstream probes, ensuring high-accuracy probe detection.

### 2.3. Sample Collection

Peripheral blood and urine samples (10 mL each) were collected from patients into Collecting Tubes (Streck, La Vista, NE, USA) within 24 h before allograft biopsy. The collected samples were stored at 4–37 °C for less than 24 h before centrifugation. The blood and urine were then individually processed via two-step centrifugation. First, samples were centrifuged at 1600× *g* for 10 min. The resulting supernatant was collected into 1.5 mL EP tubes, and the remaining precipitate was then transferred into 1.5 mL EP tubes and stored at −20 °C. Next, the collected supernatant was centrifuged at 16,000× *g* for 10 min in a low-temperature centrifuge (4 °C), and the supernatant resulting from this step was transferred into 1.5 mL EP tubes and stored at −80 °C according to our established protocol [8].

### 2.4. Peripheral Blood Genomic DNA and Urinary cfDNA Extraction and Library Construction Sequencing

Urinary cfDNA was extracted using a cfDNA extraction kit (QIAamp® Circulating Nucleic Acid Kit, 55114, Hilden, Germany). Peripheral leukocyte genomic DNA was extracted using a genomic DNA extraction kit (QIAamp DNA Mini Kit, 51304, Hilden, Germany). cfDNA (1 μL) was used for QuantiFluorTM-ST (Promega, Madison, WI, USA) quantification, and the quality was assessed using another 1 μL of cfDNA with the Agilent 2100 (PaloAlto, California, USA). Then, 30 ng of cfDNA was fragmented for constructing gDNA and urinary cfDNA libraries using the KAPA LTP Library Preparation Kit (Kapa Biosystems, Boston, MA, USA). Capture hybridization was carried out according to the manufacturer’s protocol using 454 SNP site probes for target region capture. The capture libraries were characterized using the Agilent 2100 Bioanalyzer (High Sensitivity DNA Kit, Beijing, China), then pooled and sequenced (Illumina X-ten, 15 million, PE 150 bp, San Diego, CA, USA).

### 2.5. Calculation of cfmtDNA Content

Quantitative real-time PCR (qPCR) was used to determine the cfmtDNA relative content with a template input of 1 μL of total cfDNA (0.1–2 ng) by amplifying the mitochondrial gene ND1 and the nuclear gene β-actin. Then, we calculated the cfmtDNA concentration using Formula (1). Here, to verify the qPCR method, we calculated the absolute cfmtDNA copy numbers using the linearity of the dosage-dependently constructed standard DNA plasmid solutions.
Urinary cfmtDNA concentration = urinary cfDNA concentration × (NDI/β-Actin)(1)

### 2.6. Bioinformatics Data Analysis and Relative Quantification of ddcfmtDNA

Raw sequencing data were trimmed by removing low-quality reads, adapter-contaminated reads, and duplications resulting from PCR. The reads were then aligned against the human mitochondria genome reference (GRCh38; https://www.ncbi.nlm.nih.gov/assembly/ (28 February 2019)) using BWA (http://bio-bwa.sourceforge.net (28 February 2010)). We initially filtered out reads wherein one end aligned with the nuclear genome while the other aligned with the mtDNA reference sequence, or where both ends aligned solely with the mtDNA. Meanwhile, we identified the NUMTs in our study in accordance with previous studies [9,10]. We extracted the discordant read pairs from the aligned sequence data, where one end mapped to the nuclear genome and the other to the mtDNA reference sequence. These discordant reads were then clustered based on shared orientation and proximity within 200 bp. We filtered out the clusters supported by less than five pairs of discordant reads in our primary analysis. The NUMTs located within 500 bp of both nuclear DNA and mtDNA were classified into the same NUMT entity. Then, we identified nucleotide-resolution breakpoints using split reads, with one end spanning the translocation junction. Split reads within 500 bp of discordant reads were realigned using BLAT. We subsequently analyzed the realigned reads, where one end mapped to nuclear DNA and the other end to mtDNA-derived sequences. Breakpoints were defined by at least three split reads within the same NUMT, ensuring each NUMT possessed one nuclear breakpoint and two mitochondrial breakpoints. Reads based on identified NUMTs (Appendix A) and known human NUMT mappings [9] were removed. The remaining reads were considered to originate from mtDNA. Subsequently, all polymorphic alleles were output using SNP calling via Samtools (-A-uv-t DP, AD) from the remaining reads. These SNPs were efficient sites for ddcfmtDNA quantification. The 454 SNP sites in the peripheral blood genomic DNA library were analyzed to determine the base information at each corresponding site on the recipient’s genome. The cfDNA libraries based on recipient SNP site information were analyzed to determine the proportion of heterologous SNP signals at recipient SNP sites relative to the total sequencing signals of cfDNA at those sites. The calculation method is as follows: Assume there is a valid SNP site on the recipient genome with base A and that a heterologous SNP site C is detected. The concentration of heterologous cfmtDNA at this site can then be calculated as follows:donor cell-free mtDNA (%) = donor mtDNA polymorphic site reads number/total reads number of mapping to mtDNA (2)

Based on the actual heterologous SNP signals detected at each site, the average of the remaining heterologous SNP sites can be calculated after removing outliers with a deviation from the mean exceeding twice the standard deviation, which is the sample’s relative ddcfmtDNA content.

### 2.7. Validation of Assay Precision

The ‘donor’ cfDNA was mixed with the ‘recipient’ cfDNA at concentrations of 5%, 10%, 20%, 40%, and 60%, with three replicates for each concentration gradient. Mixed cfDNA libraries were constructed as previously described [8], and a hybrid capture hybrid cfDNA library was constructed with sequencing of the target region. The proportion of ‘donor’ cfmtDNA in the mixed cfDNA libraries was then determined, and a linear regression graph was plotted with a statistical analysis based on consistency with theoretical proportions (Figure 2).

### 2.8. Statistical Analysis

Statistical analysis and visualization were conducted using GraphPad Prism (Version 9). Differences in cfmtDNA, ddcfDNA, and ddcfmtDNA levels among the three cohorts were analyzed by the Kruskal–Wallis rank–sum test. Electron microscopy images of mitochondria in allograft biopsy tissues were analyzed using ImageJ software (Version 1.52). For each individual, the cross-sectional area of 100 mitochondria in renal tubular epithelial cells was measured. A receiver operating characteristics (ROCs) analysis was performed to evaluate the diagnostic capacity of measurements in BKPyVAN. *p* < 0.05 is considered to indicate statistically significant differences.

## 3. Results

A total of 55 kidney transplant recipients were included in the study, comprising 12 STAs, 22 TCMRs, and 21 biopsy-proven BKPyVAN, as reported in our previous study [7]. The urinary cfDNA was collected concurrently with conducting the biopsy. The overall urinary cfmtDNA constituted a substantial proportion, approximately 30–40% of the total urinary cfDNA levels in kidney transplant recipients (Figure 3A). Interestingly, in kidney transplant recipients diagnosed with BKPyVAN, both the median urinary cfmtDNA concentration and fraction levels were significantly lower compared to those in patients with TCMR (3.160 vs. 3.825 ng/mL, *p* = 0.001; and 25.0 vs. 48.1%, *p* < 0.001; Figure 3A,B).

The ddcfmtDNA levels in the STA, TCMR, and BKPyVAN cohorts were further compared. The urinary ddcfmtDNA concentrations were significantly lower in the BKPyVAN cohort than in the TCMR cohort, with median levels of 0.46 and 0.57 ng/mL, respectively (*p* = 0.010, Figure 3C). In contrast, the BKPyVAN cohort exhibited significantly higher levels of urinary ddcfmtDNA (ddcfmtDNA fractions) than the STA and TCMR cohorts (*p* = 0.001 and *p* = 0.004, Figure 3D). This observation could be explained by more severe mitochondrial damage in the recipient urothelial cells of the BKPyVAN cohort. Additionally, there was a significantly lower proportion of urinary ddcfmtDNA in the total urinary ddcfDNA (ddcfmtDNA to ddcfDNA ratio) in the BKPyVAN cohort compared to the STA and TCMR cohorts (both *p* < 0.001, Figure 3E), with median levels of 3.82%, 9.50%, and 7.50%, respectively. These results are indicative of mitochondrial damage in the graft cells of the BKPyVAN cohort.

Electron microscopy was employed to visualize mitochondria in the allograft biopsy tissues (Figure 4A–C). In the STA cohort, the mitochondria are morphologically normal, being elongated with lamellar cristae, and a small proportion are short rod-shaped. Partially swelling mitochondria with reduced and disorganized cristal membranes were observed in the TCMR cohort, yet the inner boundary membranes remained visibly unaltered. Consistent with the urinary cfmtDNA measurements, we observed swelling and vacuolated mitochondria with disarrangement and densely packed cristae in the tubular cells of the BKPyVAN cohort, which supports the notion of mitochondrial dysfunction in BKPyVAN. The mitochondrial area was further measured for each cohort using ImageJ. The median mitochondrial area in the BKPyVAN cohort was 0.29 μm^2^ (Figure 4D), which is significantly lower than that in the STA (0.44 μm^2^, *p* = 0.030) and TCMR (0.48 μm^2^, *p* < 0.030) cohorts.

A receiver operating curve analysis was performed to assess the diagnostic capacity of urinary ddcfmtDNA in BKPyVAN. The relative levels of urinary ddcfmtDNA were found to contribute to BKPyVAN diagnosis, as indicated by the area under the curve (AUC = 0.778, Figure 5). Additionally, the urinary ddcfmtDNA-to-ddcfDNA ratio showed potential for distinguishing BKPyVAN from other conditions, with a cutoff value of 4.96% (AUC = 0.933, 71.4% sensitivity, and 97.1% specificity, Figure 5).

## 4. Discussion

In the present study, we demonstrated the potential for non-invasive measurements of urinary ddcfmtDNA to provide valuable insights into the diagnosis of BKPyVAN after kidney transplantation. There are three key findings resulting from our study: (1) both urinary cfmtDNA concentrations and fractions are significantly decreased in BKPyVAN; (2) in parallel with the reduced urinary cfmtDNA levels, we observed reductions in the volume of mitochondria in tubular cells in BKPyVAN; (3) the urinary ddcfmtDNA-to-ddcfDNA ratio holds promise as an upcoming non-invasive biomarker to distinguish BKPyVAN, with a cutoff value of 4.96% (with a sensitivity of 71.4% and a specificity of 97.1%).

Mitochondria are essential organelles that play key roles in thermogenesis, calcium homeostasis, and the regulation of intrinsic metabolic pathways [11]. Their significance is underscored by their involvement in various diseases, where mitochondrial damage becomes a contributing factor, resulting in the release of mtDNA into circulation, which either (1) serves as a biomarker reflecting the extent of mitochondrial damage or (2) potentially triggers a proinflammatory response. Multiple kidney diseases, including acute kidney injury, Alport syndrome, and CoQ nephropathy, are associated with mitochondrial disorders [12]. Understanding the intricate interplay between mitochondrial dysfunction and renal conditions opens avenues for exploring novel diagnostic biomarkers and therapeutic interventions in the realm of kidney-related disorders.

Decreased urinary cfmtDNA levels and ddcfmtDNA-to-ddcfDNA ratios were observed in patients with BKPyVAN compared with those of the STA and TCMR cohorts. Recent investigations have highlighted the critical role of mitochondria as organelles involved in virus infection and replication. Viruses can employ their own-encoded proteins to anchor mitochondria, aiming to exploit the host environment. Electron microscopy revealed reductions in both the quantity and volume of mitochondria in the tubular cells of the BKPyVAN cohort, supporting the notion of mitochondrial dysfunction in BKPyVAN. Consequently, the observed decrease in urinary cfmtDNA levels reflects a steep decline in mitochondria, possibly part of a defense mechanism against the immune system and impaired self-replication after BK polyomavirus infection. Notably, the majority of urinary cfmtDNA originates from urothelial cells. Therefore, the reduced urinary cfmtDNA levels in BKPyVAN are a reflection of virus reactivation and replication occurring not only in donor cells but also in recipient cells.

Due to similarities in the manifest histopathology of tubulitis and interstitial inflammation, distinguishing BKPyVAN and TCMR is challenging when SV40 staining is negative. This poses a conundrum for decision making regarding whether enhanced or reduced immunosuppression therapy should be initiated. In our previous study, we identified a noteworthy increase in both the urinary concentrations and fractions of ddcfDNA in cases of BKPyVAN compared to instances of TCMR. Consistent with our findings, Huang et al. also reported that elevated urinary ddcfDNA concentrations are associated with various types of allograft injury in kidney transplantation [13], particularly in probable and biopsy-proven BKPyVAN [14]. Aligning with these previous findings, we demonstrated that the urinary ddcfmtDNA-to-ddcfDNA ratio is a promising biomarker for distinguishing BKPyVAN from TCMR with both high sensitivity and specificity.

There are some limitations to the present study. Firstly, the relatively small sample size is attributed to the low prevalence of BKPyVAN in our center. A decreased proportion of ddcfmtDNA in total urinary ddcfDNA could also be the result of other causes, including the spread of BKPyV infection, interstitial fibrosis/tubular atrophy, acute tubular necrosis, etc. Here, we uncovered the potential for urinary ddcfmtDNA to be used to distinguish BKPyVAN from TCMR, which could have valuable clinical significance, especially when SV40 staining is negative. However, larger-scale investigations are warranted to further explore the utilization of urinary cfmtDNA as part of a non-invasive method in the diagnosis of BKPyVAN. In addition, the dynamic efficacy of monitoring urinary cfmtDNA in BKPyVAN treatment was not extensively evaluated in this study, though our ongoing research endeavors are currently focused on addressing this aspect, with the aim of facilitating a more comprehensive understanding of the role of urinary cfmtDNA in the context of BKPyVAN treatment.

## 5. Conclusions

In conclusion, we introduce a novel approach for the non-invasive monitoring of microcosmic mitochondrial damage through urinary cfmtDNA testing. The ddcfmtDNA-to-ddcfDNA ratio demonstrates superior capability in distinguishing BKPyVAN from TCMR. The understanding of mitochondrial dynamics during viral infections remains a nascent field, yet it holds promise for molecular investigations in virology. Considering the association of mitochondrial disorders with various kidney diseases, including acute kidney injury, Alport syndrome, and CoQ nephropathy, we propose extending this method for use in transplants and for potentially alerting one to the possibility of native kidney diseases. Further studies are warranted to advance this area of research.

## Figures and Tables

**Figure 1 biomolecules-14-00348-f001:**
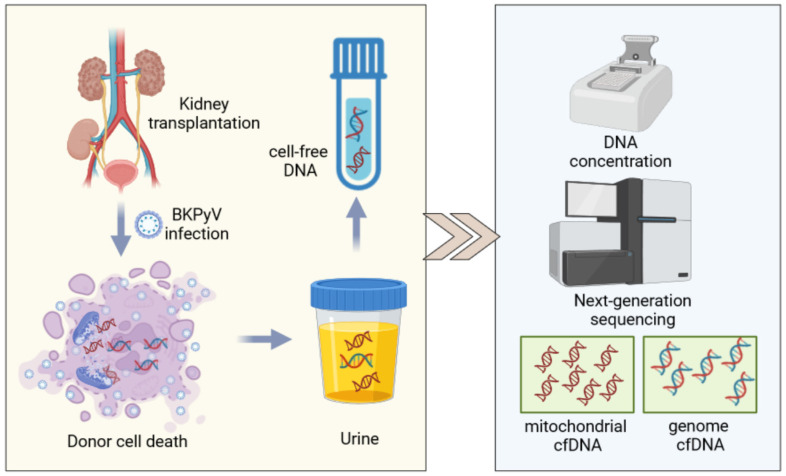
Work scheme of urinary cell-free DNA extraction. Created by BIORENDER (biorender.com).

**Figure 2 biomolecules-14-00348-f002:**
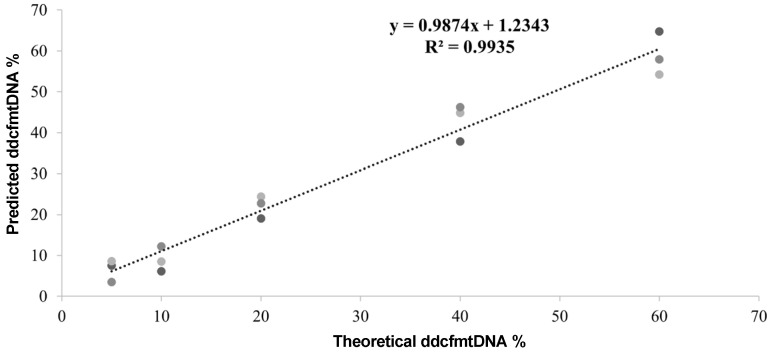
The ddcfmtDNA assay identifying ‘donor’ cfmtDNA with high linearity and accuracy. Targeted SNP capture sequencing was used to determine the percentage of ‘donor’ mtDNA present (y-axis). The x-axis represents the percentage of the theoretical mixture. The linear curve fit to these points has a slope of 0.9974 and an R^2^ value of 0.9935. The gray colored dots represent samples.

**Figure 3 biomolecules-14-00348-f003:**
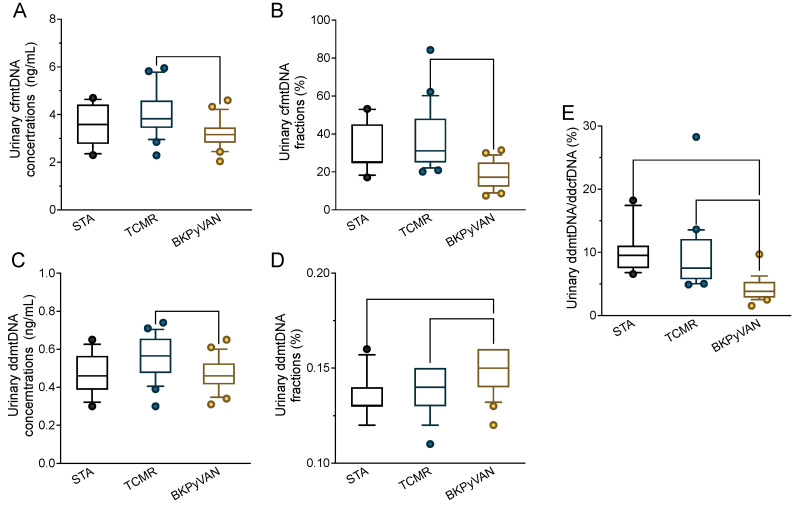
Urinary donor-derived cell-free mitochondrial DNA reflects mitochondrial damage in BKPyVAN. (**A**) Urinary cfmtDNA concentration; (**B**) urinary cfmtDNA fraction; (**C**) urinary ddcfmtDNA concentration; (**D**) urinary ddcfmtDNA-to-total cfmtDNA ratio; (**E**) urinary ddcfmtDNA-to-ddcfDNA ratio in kidney transplant recipients with STA, TCMR, and BKPyVAN. Boxplot with bold lines represents median levels: the box indicates the interquartile range, the whiskers indicate the 10–90 percentile range, and the dots represent outliers. cfmtDNA, cell-free mitochondrial DNA; ddcfmtDNA, donor-derived cfmtDNA; STA, stable graft function; TCMR, T cell-mediated rejection; BKPyVAN, BK polyomavirus-associated nephropathy.

**Figure 4 biomolecules-14-00348-f004:**
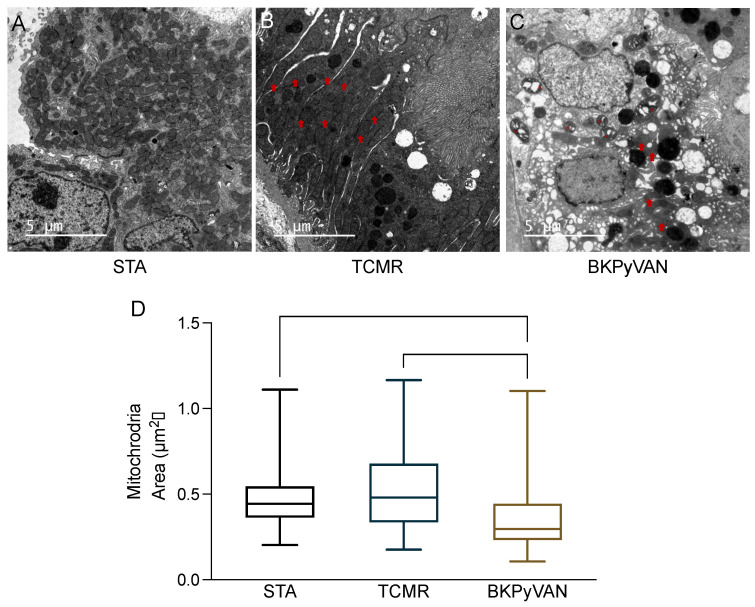
Representative electron microscopy images of mitochondria in tubular cells of kidney transplant patients. (**A**) Normal, elongated mitochondria with lamellar cristae, and a small portion of mitochondria are short rod-shaped in the STA cohort; (**B**) swelling mitochondria with reduced and disorganized cristal membranes in the TCMR cohort; and (**C**) swelled and vacuolated mitochondria with disarrangement and densely packed cristae in the BKPyVAN cohort; scale = 5 μm. (**D**) Mitochondrial area in STA, TCMR, and BKPyVAN cohorts. Violin plot with dashed line represents median levels, and dotted line represents quantiles. STA, stable graft function; TCMR, T-cell mediate rejection; BKPyVAN, BK polyomavirus-associated nephropathy. Arrows indicate swelling mitochondria, and * indicates vacuolated mitochondria.

**Figure 5 biomolecules-14-00348-f005:**
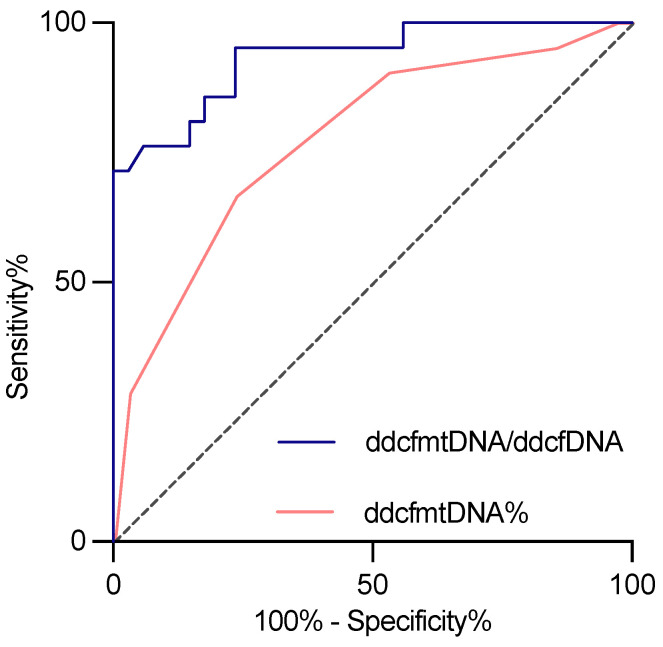
Receiver operating curve of urinary ddcfmtDNA/ddcfDNA (blue) and ddcfmtDNA% (red) to distinguish BKPyVAN.

## Data Availability

All data and materials included in this study are available upon request by contacting the corresponding author.

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
