# Peer review of "Cell-Free Mitochondrial DNA: An Upcoming Non-Invasive Tool for Diagnosis of BK Polyomavirus-Associated Nephropathy"

_biomolecules, 2024, doi:10.3390/biom14030348_

Round 1
Reviewer 1 Report
Comments and Suggestions for Authors
Manuscript by Guo et al. (biomolecules-2855317) examined potential of non-invasively monitoring in kidney diseases using urinary donor-derived cell-free mitochondrial DNA (ddcfmtDNA). The two major causes of kidney failure after transplantation are BK polyomavirus-associated nephropathy (BKPyVAN) and T cell-mediated rejection (TCMR). Distinguish between BKPyVAN and TCMR is vital because their treatment directions are quite opposite. Cell-free DNA (cfDNA) has been reported in several literature as one of criteria for diagnosis of BKPyVAN including authors of the current manuscript (5,7,11). They described here the use of ddcfmtDNA could applied as diagnostic measure for BKPyVAN. Results were somewhat confusing. Total amount of urine cfmtDNA of BKPyVAN was significantly lower than those of TCMR, but proportion of donor derived cfmtDNA (among total cfmtDNA) was significantly higher in BKPyVAN patients. The interpretation of the results would be that mitochondrial DNA release in BKPyVAN patient was restricted to BKPyV infected cells, but TCMR damages could be more global on transplanted kidney tissue. The ratio of ddcfmtDNA among total ddcfDNA in BKPyVAN showed significantly lower value than others suggesting that such criteria could be used to distinguish BKPyVAN from TCMR. The authors appeared to be one of research groups actively involved in clinical research area in Urology. The current manuscript described interesting approach on BKPyVAN diagnosis, though this reviewer feels that the authors should conduct more thorough research to draw conclusions.
Major points
1. The criteria that decreased proportion of ddcfmtDNA among total ddcfDNA could be tricky in some possible situations. What if the BKPyV infection was more spread than those the current manuscript dealt with? It could create more ddcfDNA and more ddcfmtDNA that may contradict the authors’ conclusions. Multiple sampling points could resolve such complications and would reveal more detailed view what aspects the cfDNAs would represent. As addressed in the manuscript more data will be needed to convince readers.
2. How to quantify cfmDNA and ddcfmDNA was not clear. Was it from read count ratio multiplied by total cfDNA amount? Methods described deep sequencing analyses of cfDNA to extract cfmDNA value was somewhat incomplete. It should be concisely described as in Reference 8.
3. They claimed that ”reductions in both quantity and volume of mitochondria in tubular cells in BKPyVAN” based on observation under EM, though they require quantitative measure to support their conclusion.
Minor point
Line 126; What is “(ddcfmtDNA to ddcfmtDNA ratio)”?
Author Response
#Reviewer 1
Manuscript by Guo et al. (biomolecules-2855317) examined potential of non-invasively monitoring in kidney diseases using urinary donor-derived cell-free mitochondrial DNA (ddcfmtDNA). The two major causes of kidney failure after transplantation are BK polyomavirus-associated nephropathy (BKPyVAN) and T cell-mediated rejection (TCMR). Distinguish between BKPyVAN and TCMR is vital because their treatment directions are quite opposite. Cell-free DNA (cfDNA) has been reported in several literature as one of criteria for diagnosis of BKPyVAN including authors of the current manuscript (5,7,11). They described here the use of ddcfmtDNA could applied as diagnostic measure for BKPyVAN. Results were somewhat confusing. Total amount of urine cfmtDNA of BKPyVAN was significantly lower than those of TCMR, but proportion of donor derived cfmtDNA (among total cfmtDNA) was significantly higher in BKPyVAN patients. The interpretation of the results would be that mitochondrial DNA release in BKPyVAN patient was restricted to BKPyV infected cells, but TCMR damages could be more global on transplanted kidney tissue. The ratio of ddcfmtDNA among total ddcfDNA in BKPyVAN showed significantly lower value than others suggesting that such criteria could be used to distinguish BKPyVAN from TCMR. The authors appeared to be one of research groups actively involved in clinical research area in Urology. The current manuscript described interesting approach on BKPyVAN diagnosis, though this reviewer feels that the authors should conduct more thorough research to draw conclusions.
Thank you for your constructive comments to help improve the quality of the manuscript.
Major points
- The criteria that decreased proportion of ddcfmtDNA among total ddcfDNA could be tricky in some possible situations. What if the BKPyV infection was more spread than those the current manuscript dealt with? It could create more ddcfDNA and more ddcfmtDNA that may contradict the authors’ conclusions. Multiple sampling points could resolve such complications and would reveal more detailed view what aspects the cfDNAs would represent. As addressed in the manuscript more data will be needed to convince readers.
Thank you for kindly reminding us this limitation. Decreased proportion of ddcfmtDNA in total ddcfDNA could result from some possible situations including spread BKPyV infection, IFTA, ATN and ect. We here in uncovered the potential usage of urinary ddcfmtDNA in distinguish BKPyVAN from TCMR, which could provide valuable clinical significance especially when SV40 staining is negative. We hope that this measurement could guide precise management in such circumstance. However, the prevalence rate of BKPyVAN (approximately 2%) in our kidney disease center is relatively low because of effective clearance of BKPyV infection due to practical experiment and routine surveillance of urinary BKPyV loads. We were only able to enroll 21 biopsy-proven BKPyVANs in our current study. Larger scale study and dynamic monitoring would reveal more detailed view what aspects the cfDNAs would represent. We have taken your suggestions into account added this into limitation.
We have added this limitation into the Discussion sections,
Line 270 as,
“A decreased proportion of ddcfmtDNA in total urinary ddcfDNA could also be the result of other causes, including spreading of BKPyV infection, interstitial fibrosis/tubular at-rophy, acute tubular necrosis, etc. Here, we uncovered the potential for urinary ddcfmtDNA to be used in distinguishing BKPyVAN from TCMR, which could have valuable clinical significance especially when SV40 staining is negative.”
- How to quantify cfmDNA and ddcfmDNA was not clear. Was it from read count ratio multiplied by total cfDNA amount? Methods described deep sequencing analyses of cfDNA to extract cfmDNA value was somewhat incomplete. It should be concisely described as in Reference 8.
Sure! Detailed methods have been added in the manuscript. Line 98 as,
"2.4. Peripheral blood Genomic DNA and Urinary cfDNA Extraction and Library Con-struction Sequencing Urinary cfDNA was extracted using a cfDNA extraction kit (QIAamp® Circulating Nucleic Acid Kit, 55114). Peripheral leukocyte genomic DNA was extracted using a genomic DNA extraction kit (QIAamp DNA Mini Kit, 51304). cfDNA (1 μl) was used for QuantiFluorTM-ST (Promega) quantification, and the quality was assessed using another 1 μl of cfDNA with the Agilent 2100.……Based on the actual heterologous SNP signals detected at each site, the average of the remaining heterologous SNP sites can be calculated after removing outliers with deviation from the mean exceeding twice the standard deviation, which is the sample's relative ddcfmtDNA content.”
- They claimed that ”reductions in both quantity and volume of mitochondria in tubular cells in BKPyVAN” based on observation under EM, though they require quantitative measure to support their conclusion.
Thank you for your constructive comments to help improve the quality of the manuscript. The mitochondrial area under EM in each cohort was measured by ImageJ, and we found the median mitochondrial area in the BKPyVAN cohort was significantly lower than the STA and TCMR cohorts. We have revised this in the Results and Discussion sections,
Line 193 as,
“In the STA cohort, the mitochondria are morphologically normal, being elongated with lamellar cristae, and a small proportion are short rod-shaped. Partially swelling mitochondria with reduced and disorganized cristal membranes were observed in the TCMR cohort, yet the inner boundary membranes remained visibly unaltered. Consistent with the urinary cfmtDNA measurements, we observed swelling and vacuolated mitochondria with disarrangement and densely packed cristae in tubular cells of BKPyVAN, which supports the notion of mitochondrial dysfunction in BKPyVAN. The mitochondrial area was further measured for each cohort using ImageJ. The median mitochondrial area in the BKPyVAN cohort was 0.29μm2 (Fig. 4D), which is significantly lower than in the STA (0.44μm2, p=0.030) and TCMR (0.48μm2, p<0.030) cohorts.”
And line 231 as,
“in parallel with the reduced urinary cfmtDNA levels, we observed reductions in volume of mitochondria in tubular cells in BKPyVAN.”
Minor point
Line 126; What is “(ddcfmtDNA to ddcfmtDNA ratio)”?
We apologize this as typo, it should be “ddcfmtDNA to ddcfDNA ratio” and we have already corrected it. Thank you! See Line 188.
Reviewer 2 Report
Comments and Suggestions for Authors
Major concerns:
Materials and methods:
1. Authors need to describe how did they measure absolute concentration of mtDNA in the sequence’s samples and how did they distinguish mtDNA and nuclear mitochondrial (NUMTs) sequences in urine.
2. It is not clear why authors have not used “Validation of assay precision for dd-cfmtDNA” which they have performed in their previous publication for ddcfDNA (Ref.8).
3. Authors need to describe how did they select the candidate for mtDNA SNPs.
Results:
Fig.3: Mitochondria need to be indicated with arrows/arrowheads in each panel. How did authors define “damaged” mitochondria on BKPyVAN panel? They mentioned “we observed reductions in both quantity and volume of mitochondria in tubular cells in BKPyVAN”(lanes 157-158) and “reductions in both the quantity and volume of mitochondria in tubular cells BKPyVAN (lanes 176-177). I think it requires additional clarification and better description of mitochondrial morphology, with explanation of parameters of intact and damaged mitochondria, including cristae, mitochondrial membrane, and others (e.g. aspect ratio).
Minor:
Lane 19 and further in the text-it is “mitochondrial DNA”.
Comments on the Quality of English LanguageMinor editing of English language required and minor typos need to be checked through the text
Author Response
#Reviewer 2
Major concerns:
Materials and methods:
- Authors need to describe how did they measure absolute concentration of mtDNA in the sequence’s samples and how did they distinguish mtDNA and nuclear mitochondrial (NUMTs) sequences in urine.
Sure! Detailed methods have been added in the manuscript. Line 118 as,
“Raw sequencing data were trimmed by removing low-quality reads, adapt-er-contaminated reads, and duplications resulting from PCR. The reads were then aligned against the human mitochondria genome reference (GRCh38; https://www.ncbi. nlm.nih.gov/assembly/) using BWA (http://bio-bwa.sourceforge.net). All polymorphic alleles were output using SNP calling via Samtools (-A -uv-t DP, AD). Sites found to correspond to nuclear mitochondrial DNA segments (NUMTs) based on the human NUMTs map were removed [9]. The 454 SNP sites in the peripheral blood genomic DNA library were analyzed to determine the base information at each corresponding site on the recipient's genome.”
- It is not clear why authors have not used “Validation of assay precision for dd-cfmtDNA” which they have performed in their previous publication for ddcfDNA (Ref.8).
Thank you for your constructive comments to help improve the quality of the manuscript. We have added “Validation of assay precision for dd-cfmtDNA” in Methods section. Line 138 as,
“2.7. Validation of assay precision
The ‘donor’ cfDNA was mixed with ‘recipient’ cfDNA at concentrations of 5%, 10%, 20%, 40%, and 60%, with three replicates for each concentration gradient. Mixed cfDNA libraries were constructed as previously described [8], and a hybrid capture hybrid cfDNA library constructed with sequencing of the target region. The proportion of ‘donor’ cfmtDNA in mixed cfDNA libraries was then determined and a linear regression graph plotted with statistical analysis based on consistency with theoretical proportions (Fig. 2).”
- Authors need to describe how did they select the candidate for mtDNA SNPs.
Sure! Details about candidate selection of mtDNA SNPs have been added in the manuscript. Line 77 as,
“2.2. Single nucleotide polymorphism (SNP) site selection and probe synthesis
SNP sites were selected from dbSNP (https://www.ncbi.nlm.nih.gov/snp/) and the Chinese population database (maintained in AlloDx Biotech, Co. Ltd. Shanghai), which is a catalog of genetic polymorphism sites constructed based on the whole-genome sample information of 5200 Chinese individuals. In total, 454 target SNP loci were obtained by screening for sites with minor allele frequencies (MAF) of close to 0.45-0.55. Based on the genomic coordinates of these 454 target SNP sites, we designed upstream and downstream probes by extending 80bp nucleotide sequences from the SNP site coordinates for both the positive and negative strands, excluding the SNP site sequences. Bidirectional coverage detection of each SNP site was achieved by using both upstream and downstream probes, ensuring high accuracy of probe detection.”
Results:
Fig.3: Mitochondria need to be indicated with arrows/arrowheads in each panel. How did authors define “damaged” mitochondria on BKPyVAN panel? They mentioned “we observed reductions in both quantity and volume of mitochondria in tubular cells in BKPyVAN”(lanes 157-158) and “reductions in both the quantity and volume of mitochondria in tubular cells BKPyVAN (lanes 176-177). I think it requires additional clarification and better description of mitochondrial morphology, with explanation of parameters of intact and damaged mitochondria, including cristae, mitochondrial membrane, and others (e.g. aspect ratio).
Thank you for your insightful comments. We have taken your suggestions into account and made the necessary revisions. We have rewritten the description of mitochondrial morphology, and we really believe our manuscript has been significantly improved.
Line 193 as,
“In the STA cohort, the mitochondria are morphologically normal, being elongated with lamellar cristae, and a small proportion are short rod-shaped. Partially swelling mitochondria with reduced and disorganized cristal membranes were observed in the TCMR cohort, yet the inner boundary membranes remained visibly unaltered. Consistent with the urinary cfmtDNA measurements, we observed swelling and vacuolated mitochondria with disarrangement and densely packed cristae in tubular cells of BKPyVAN, which supports the notion of mitochondrial dysfunction in BKPyVAN. The mitochondrial area was further measured for each cohort using ImageJ. The median mitochondrial area in the BKPyVAN cohort was 0.29μm2 (Fig. 4D), which is significantly lower than in the STA (0.44μm2, p=0.030) and TCMR (0.48μm2, p<0.030) cohorts.”
And line 207 as,
“(A) Normal, elongated with lamellar cristae, and a small portion of mitochondria are short rod-shaped in the STA cohort, (B) swelling mitochondria with reduced and disorganized cristal membranes in the TCMR cohort, and (C) swelled and vacuolated mitochondria with disarrangement and densely packed cristae in the BKPyVAN cohort, scale=5μm. (D) Mitochondrial area in STA, TCMR, and BKPyVAN cohorts. Violin plot with dashed line representing median levels and dotted line representing quantiles. STA, stable graft function; TCMR, T-cell mediate rejection; BKPyVAN, BK polyomavirus-associated nephropathy. Arrows indicate swelling mitochondria and * indicates vacuolated mito-chondria.”
Minor:
Lane 19 and further in the text-it is “mitochondrial DNA”.
Thank you for your valuable comments and suggestions. We have corrected the typo.
Comments on the Quality of English Language
Minor editing of English language required and minor typos need to be checked through the text
Thank you. We have enlisted the assistance of a professional English editing service of mdpi to refine the language, and we are confident that the manuscript has been significantly improved.
Reviewer 3 Report
Comments and Suggestions for Authors
Manuscript ID: biomolecules-2855317
The main aim of the study was to develop a method for diagnosis of BK polyomavirus-associated nephropathy (BKPyVAN), which is based on noninvasive monitoring of mitochondrial impairment through urinary donor derived cell-free mitochondrial DNA (ddcfmtDNA) analysis. The research was conducted on urine samples from 60 kidney transplant recipients, which were classified into three groups: stable graft function (n=12), T cell-mediated rejection (n=22), and biopsy-proven BKPyVAN (n=21). The main finding of the study was that ddcfmtDNA to ddcfDNA ratio is a marker that can be used to distinguish BKPyVAN from other conditions, with a cutoff 4.96%, AUC 0.93, sensitivity 71% and specificity 97%. Elevated ddcfmtDNA is a sign of mitochondrial damage, which was visualized with electron microscopy. This method could be used for monitoring mitochondrial damage related to BKPyVAN.
Please correct or explain the following issues:
1. Sample collection – please explain the wide temperature range of sample storage before centrifugation.
2. Details or reference on DNA sequencing are missing and should be provided.
3. Statistical analysis – explain what kind of data is presented in Figure 2, mean, median? Please explain what is represented by boxes, whiskers and dots. Exact p-values should be given.
4. Please explain how do you define urinary cfmtDNA fraction.
5. Minor corrections:
· Page 2, line 80 - word 'were' is missing.
· Page 2 line 48 - 'Julia et al.' should be replaced with the surname of the cited author.
· Increase the font in axis titles of Figure 2.
Author Response
#Reviewer 3
The main aim of the study was to develop a method for diagnosis of BK polyomavirus-associated nephropathy (BKPyVAN), which is based on noninvasive monitoring of mitochondrial impairment through urinary donor derived cell-free mitochondrial DNA (ddcfmtDNA) analysis. The research was conducted on urine samples from 60 kidney transplant recipients, which were classified into three groups: stable graft function (n=12), T cell-mediated rejection (n=22), and biopsy-proven BKPyVAN (n=21). The main finding of the study was that ddcfmtDNA to ddcfDNA ratio is a marker that can be used to distinguish BKPyVAN from other conditions, with a cutoff 4.96%, AUC 0.93, sensitivity 71% and specificity 97%. Elevated ddcfmtDNA is a sign of mitochondrial damage, which was visualized with electron microscopy. This method could be used for monitoring mitochondrial damage related to BKPyVAN.
Thank you for your constructive comments to help improve the quality of the manuscript.
Please correct or explain the following issues:
- Sample collection – please explain the wide temperature range of sample storage before centrifugation.
The preservation solution in cfDNA preservation tubes prevents cfDNA degradation, preserves cell integrity, and avoids the contaminate of newly released cfDNA during transportation; this preservation solution has stable properties between 4-37℃ and is suitable for most transportation conditions.
- Details or reference on DNA sequencing are missing and should be provided.
Sure! Detailed methods have been added in the manuscript. Line 98 as,
“2.4. Peripheral blood Genomic DNA and Urinary cfDNA Extraction and Library Con-struction Sequencing Urinary cfDNA was extracted using a cfDNA extraction kit (QIAamp® Circulating Nucleic Acid Kit, 55114). Peripheral leukocyte genomic DNA was extracted using a genomic DNA extraction kit (QIAamp DNA Mini Kit, 51304).……Based on the actual heterologous SNP signals detected at each site, the average of the remaining heterologous SNP sites can be calculated after removing outliers with deviation from the mean exceeding twice the standard deviation, which is the sample's relative ddcfmtDNA content.”
- Statistical analysis – explain what kind of data is presented in Figure 2, mean, median? Please explain what is represented by boxes, whiskers and dots. Exact p-values should be given.
Thank you for kindly reminding us this drawback. The data presented in Figure 2 was boxplot with bold line representing median levels, box indicating the interquartile range, whiskers indicating 10-90 percentile range and dots representing outliers. And the exact p-values were given in the manuscript.
We have revised the manuscript, see line 175. Highlighted in yellow.
“Boxplot with bold lines representing median levels, the box indicating the interquartile range, whiskers indicating the 10-90 percentile range, and dots representing outliers. cfmtDNA, cell-free mitochondrial DNA.”
- Please explain how do you define urinary cfmtDNA fraction.
Sure!Quantitative real-time PCR was used to determined cfmtDNA content (urinary cfmtDNA fraction) with a template input of 1 μl total cfDNA (0.1-2ng), by amplificated the mitochondrial gene ND1 and the nuclear gene β-actin. Urinary cfmtDNA fraction is defined as NDI/ β- Actin.
- Minor corrections:
- Page 2, line 80 - word 'were' is missing.
- Page 2 line 48 - 'Julia et al.' should be replaced with the surname of the cited author.
- Increase the font in axis titles of Figure 2.
Thank you! These mistakes have been corrected.
Round 2
Reviewer 1 Report
Comments and Suggestions for Authors
The revised manuscript by Guo et al. (biomolecules-2855317) examined potential of non-invasive diagnosis of BK polyomavirus-associated nephlopathy using urinary donor-derived cell-free mitochondrial DNA (ddcfmtDNA). The text appeared greatly improved after the revision; questions raised by reviewers appeared properly answered with additional data, ensuring the manuscript now meets the required standards for publication. This reviewer is of the opinion that the current version of the manuscript is well-suited for publication in Biomolecules.
Author Response
Thank you for your appreciation.
Reviewer 2 Report
Comments and Suggestions for Authors
Major concerns:
1.Lane 110-“Calculation of cfmtDNA content”- needs additional clarification. It is not clear to me why authors used the formula (1), if they calculated cf-mtDNA concentration using qPCR with standard calibration curve with ND1-specific standards? The formula (1) is not clear to me anyway.
2.Lanes 122-123-“Sites found to correspond to nuclear mitochondrial DNA segments (NUMTs) based on the human NUMTs map were removed [9].” Authors cited Ref.9, which has actually described a few previously proposed pipelines/algorithms for identification of NUMTs in human mtDNA. Authors should provide detailed protocol for identification of NUMTs in their study. Although NUMTs is not the major object of the study, it will be very informative if authors add the part about how many NUMTs sites, including size and coverage they identified.
3. Lane 199: “The mitochondrial area was further measured for each cohort using ImageJ”- Mitochondria area/size evaluation needs better description, e,i. how many images were acquired per group? How many mitochondria were calculated? My suggestion regarding demonstrating of data in violin plots to an audience unfamiliar with the violin plot- it might be better to go with a simpler and more straightforward visualization like the box plot.
Author Response
1.Lane 110-“Calculation of cfmtDNA content”- needs additional clarification. It is not clear to me why authors used the formula (1), if they calculated cf-mtDNA concentration using qPCR with standard calibration curve with ND1-specific standards? The formula (1) is not clear to me anyway.
We really appreciate your comments. The standard calibration curve was used for verifying the qPCR method. The detailed description seen, Line 112 as,
“Quantitative real-time PCR (qPCR) was used to determined cfmtDNA relative content with a template input of 1 μl total cfDNA (0.1-2ng), by amplificated the mitochondrial gene ND1 and the nuclear geneβ-actin. Then we calculated the cfmtDNA concentration using the formula (1) . Here, to verify the qPCR method, we calculated the absolute cfmtDNA copy numbers using the linearity of the dosage-dependently constructed standard DNA plasmid solutions.
Urinary cfmtDNA concentration = urinary cfDNA concentration * (NDI/β-Actin) (1)”
2.Lanes 122-123-“Sites found to correspond to nuclear mitochondrial DNA segments (NUMTs) based on the human NUMTs map were removed [9].” Authors cited Ref.9, which has actually described a few previously proposed pipelines/algorithms for identification of NUMTs in human mtDNA. Authors should provide detailed protocol for identification of NUMTs in their study. Although NUMTs is not the major object of the study, it will be very informative if authors add the part about how many NUMTs sites, including size and coverage they identified.
Thank you for your insightful comments to improve our manuscript. Detailed description of NUMTs identification has been included in Method section, Line 123 as,
“We initially filtered out reads wherein one end aligned with the nuclear genome while the other aligned with the mtDNA reference sequence, or where both ends aligned solely with the mtDNA. Meanwhile, we identified the NUMTs in our study in accordance with previous studies [9,10] . We extracted the discordant read pairs from the aligned sequence data, where one end mapped to the nuclear genome and the other to the mtDNA ref-erence sequence. These discordant reads were then clustered based on shared orientation and proximity within 200 bp. We filtered out the clusters supported by less than five pairs of discordant reads in our primary analysis. The NUMTs located within 500 bp of both nuclear DNA and mtDNA were classified into the same NUMT entity. Then we identified nucleotide-resolution breakpoints using split reads, with one end spanning the translo-cation junction. Split reads within 500 bp of discordant reads were realigned using BLAT. We subsequently analyzed the realigned reads where one end mapped to nuclear DNA and the other end to mtDNA-derived sequences. Breakpoints were defined by at least three split reads within the same NUMT, ensuring each NUMT possessed one nuclear breakpoint and two mitochondrial breakpoints. Reads based on identified NUMTs (Supplementary Table 1) and known human NUMT mappings [9] were removed. The remaining reads were considered as originating from mtDNA. Subsequently, all pol-ymorphic alleles were output using SNP calling via Samtools (-A -uv-t DP, AD) from the remaining reads. These SNPs were efficient sites for ddcfmtDNA quantity.”
- Lane 199: “The mitochondrial area was further measured for each cohort using ImageJ”- Mitochondria area/size evaluation needs better description, e,i. how many images were acquired per group? How many mitochondria were calculated? My suggestion regarding demonstrating of data in violin plots to an audience unfamiliar with the violin plot- it might be better to go with a simpler and more straightforward visualization like the box plot.
Sure! For each individual, the cross-sectional area of 100 mitochondria in renal tubular epithelial cells was measured. The violin plot is replaced by boxplot. We have revised the manuscript, see line 172 as,
“Electron microscopy images of mitochondria in allograft biopsy tissues were analyzed using ImageJ software. For each individual, the cross-sectional area of 100 mitochondria in renal tubular epithelial cells was measured.”